# Synergistic Reduction of Arsenic Uptake and Alleviation of Leaf Arsenic Toxicity in Maize (*Zea mays* L.) by Arbuscular Mycorrhizal Fungi (AMF) and Exogenous Iron through Antioxidant Activity

**DOI:** 10.3390/jof9060677

**Published:** 2023-06-15

**Authors:** Hong-Yin Zhou, Fu-Zhao Nian, Bao-Dong Chen, Yong-Guan Zhu, Xian-Rong Yue, Nai-Ming Zhang, Yun-Sheng Xia

**Affiliations:** 1College of Resources and Environment, Yunnan Agricultural University, Kunming 650201, China; zhy1605202632@163.com (H.-Y.Z.); zhangnaiming@sina.com (N.-M.Z.); 2College of Plant Protection, Yunnan Agricultural University, Kunming 650201, China; 3College of Tobacco Science, Yunnan Agricultural University, Kunming 650201, China; fuzhaonian@126.com; 4State Key Laboratory of Urban and Regional Ecology, Research Center for Eco-Environmental Sciences, Chinese Academy of Sciences, Beijing 100085, China; bdchen@rcees.ac.cn (B.-D.C.); ygzhu@rcees.ac.cn (Y.-G.Z.); 5University of Chinese Academy of Sciences, Beijing 100049, China; 6College of Marxism, Yunnan Agricultural University, Kunming 650201, China; yxr82@163.com

**Keywords:** AMF inoculation, arsenic stress, iron compounds, physiological and biochemical mechanism

## Abstract

Arbuscular mycorrhizal fungi (AMF) play key roles in enhancing plant tolerance to heavy metals, and iron (Fe) compounds can reduce the bioavailability of arsenic (As) in soil, thereby alleviating As toxicity. However, there have been limited studies of the synergistic antioxidant mechanisms of AMF (*Funneliformis mosseae*) and Fe compounds in the alleviation of As toxicity on leaves of maize (*Zea mays* L.) with low and moderate As contamination. In this study, a pot experiment was conducted with different concentrations of As (0, 25, 50 mgꞏkg^−1^) and Fe (0, 50 mgꞏkg^−1^) and AMF treatments. Results showed that under low and moderate As concentrations (As25 and As50), the co-inoculation of AMF and Fe compound significantly increased the biomass of maize stems and roots, phosphorus (P) concentration, and P-to-As uptake ratio. Moreover, the co-inoculation of AMF and Fe compound addition significantly reduced the As concentration in stem and root, malondialdehyde (MDA) content in leaf, and soluble protein and non-protein thiol (NPT) contents in leaf of maize under As25 and As50 treatments. In addition, co-inoculation with AMF and Fe compound addition significantly increased the activities of catalase (CAT), peroxidase (POD), and superoxide dismutase (SOD) in the leaves of maize under As25 treatment. Correlation analysis showed that stem biomass and leaf MDA content were very significantly negatively correlated with stem As content, respectively. In conclusion, the results indicated that the co-inoculation of AMF and Fe compound addition can inhibit As uptake and promote P uptake by maize under low and moderate As contamination, thereby mitigating the lipid peroxidation on maize leaves and reducing As toxicity by enhancing the activities of antioxidant enzymes under low As contamination. These findings provide a theoretical basis for the application of AMF and Fe compounds in the restoration of cropland soil contaminated with low and moderate As.

## 1. Introduction

Arsenic (As) is commonly present in nature and is recognized as a carcinogen and environmental pollutant [1,2,3]. Arsenic in soil can accumulate in crops and migrate into lake water and groundwater, affecting crop yield and also causing serious harm to human health and local ecosystems [4,5,6]. When crops absorb excessive amounts of arsenic from the soil, they are subject to arsenic toxicity, which can manifest as inhibited root growth, stunted plant growth and development, and even death, resulting in decreased crop yield and affecting food security [7,8,9]. Physiologically, the toxicity of arsenic leads to the accumulation of reactive oxygen species in plant cells [10]. When the reactive oxygen species produced exceed the capacity of the reactive oxygen clearance system, plants experience inhibited chlorophyll synthesis, membrane lipid peroxidation, and damage to DNA, proteins, and some biomolecules [11].

The concentration of arsenic in agricultural soils in South and East Asia has been reported to range from 0 to 6402 mg·kg^−1^, indicating severe arsenic pollution [12,13]. Maize (*Zea mays* L.) is the most widely cultivated cereal worldwide, however, a study has shown that in countries with high maize yields, such as China, Argentina, India, and Mexico, the soil arsenic concentration greatly exceeds the global average soil background value (10.0 mg·kg^−1^) [14]. Therefore, the management of arsenic-polluted soil on farmland has become an important focus in environmental science [15,16,17,18].

Plants have evolved a series of defense mechanisms to resist external environmental stressors. When plants are subjected to arsenic stress, enzymes that synthesize plant chelators (PCs) can be activated. The thiol group on PCs can form a complex with reduced trivalent arsenic, which can be transported into the vacuole for storage to alleviate arsenic toxicity. However, the synthesis of PCs consumes the plant antioxidant glutathione, thus reducing the availability of glutathione to remove reactive oxygen species [19]. When plants are subjected to oxidative stress caused by excessive reactive oxygen species, they activate their non-enzymatic antioxidant system (including glutathione, ascorbic acid, carotenoids, and soluble proteins) and enzymatic antioxidant system (including SOD, POD, CAT, glutathione reductase, and glutathione-S-transferase) [20,21]. However, when the amount of arsenic entering the cell exceeds a certain limit, the activities of the components of these antioxidant enzyme and non-enzymatic systems will be inhibited, thereby causing harm to plant cells. Previous studies have shown that soil arsenic bioavailability and plant toxicity are related to the concentration and form of arsenic in the soil, plant species, and soil properties such as the content of iron oxides, redox potential, pH value, and phosphorus (P) content [22,23].

Numerous studies [24,25,26,27] have indicated that the inoculation of arbuscular mycorrhizal fungi (AMF) can have significant effects on plants, including increased absorption and utilization of mineral nutrients, altered uptake and transport of heavy metals, and alleviation of the adverse effects of heavy metal stress, thereby helping to improve the tolerance of host plants to heavy metals. Specifically, AMF colonization can enhance the arsenic resistance of either resistant or non-resistant plants grown in arsenic-contaminated soils by reducing arsenic biotoxicity [28,29,30]. AMF colonization can effectively enhance arsenic extraction for phytoremediation of heavily polluted soils, as demonstrated for the hyperaccumulator fern plant *Pteris vittata* [31,32]. Additionally, some studies have shown that there is a high affinity between iron and arsenic in soil [33,34,35,36]. Arsenic entering the soil can be adsorbed specifically or non-specifically onto the surface of iron oxides or hydroxides in the soil to form insoluble precipitates to alleviate the toxicity of arsenic to plants.

In recent years, both domestic and international researchers have begun to investigate the synergistic effects of AMF and iron compounds and the effects on arsenic-contaminated plants [37]. Using a pot experiment, our research group [38] studied the combined effects of AMF and iron tailings to enhance plant resistance to arsenic by increasing the absorption of phosphorus and iron under moderate arsenic stress. However, there have been no reports on how the synergistic action of AMF and iron compounds alters the physiological and biochemical resistance mechanisms of leaf tissues. To address this, we employed a pot experiment to investigate the influences of different dosages of exogenous iron and AMF inoculation on physiological and biochemical indicators of maize leaves in soils contaminated with varying degrees of arsenic. The aim of this work was to elucidate the physiological and biochemical mechanisms by which AMF inoculation and exogenous iron compounds mitigate the phytotoxic effects of arsenic contamination. Specifically, we aimed to determine: (1) whether AMF and iron can synergistically regulate arsenic uptake in maize plants; (2) whether AMF and iron can synergistically alleviate the degree of membrane lipid peroxidation and reduce non-enzymatic antioxidant content of maize leaves and synergistically enhance antioxidant enzyme activities in maize leaves under arsenic stress; and (3) the plant response to different concentrations of AMF and iron under arsenic stress.

We hypothesized that (1) either AMF or iron application alone would promote maize growth and increase phosphorus uptake compared to the control; and (2) AMF and iron would synergistically alleviate the degree of membrane lipid peroxidation, reduce the non-enzymatic antioxidant content of maize leaves, and synergistically enhance antioxidant enzyme activity in maize leaves under arsenic stress.

## 2. Materials and Methods

### 2.1. Materials

The soil used in this study was a low-phosphorus sandy soil planted with tobacco from Daxing County, with the following basic physicochemical properties: pH of 7.83, organic matter content of 9.1 g·kg^−1^, cation exchange capacity of 9.5 cmol·kg^−1^, total arsenic content of 10.3 mg·kg^−1^, and total phosphorus, manganese, copper, and zinc contents of 511.0, 437.0, 19.0, and 64.0 mg·kg^−1^, respectively. The total iron (FeO) content was 2.9%, while the available arsenic and phosphorus contents were 0.3 and 5.0 mg·kg^−1^, respectively. The available iron, manganese, copper, and zinc contents were 8.8, 13.2, 1.1, and 1.0 mg·kg^−1^. The soil was sieved through a 2 mm mesh and sterilized by irradiation (20 KGy).

Maize (*Zea mays* L. ND108) was used in this study. Uniformly sized and plump maize seeds were selected, disinfected with 10% hydrogen peroxide for 10 min, and germinated for two days and two nights until whitening occurred before sowing.

*Funneliformis mosseae* BGCXJ03A was used as the AMF in this study and was provided by the Microbial Laboratory of Plant Nutrition and Resources Research Institute, Beijing Academy of Agricultural and Forestry Sciences. The fungi were isolated from the rice rhizosphere in Aksu, Xinjiang, and propagated with sorghum. Each 20 mL of the spore suspension contained 656 spores. For the AMF treatment, 50 g of AMF inoculum was mixed evenly into each pot. For the non-mycorrhizal (NM) treatment, an equal amount of sterilized inoculum and 10 mL of sterilized inoculum filtrate were mixed.

FeSO_4_·7H_2_O (analytical grade, CAS No. 7782-63-0) was used as the iron source in this study.

Na_3_AsO_4_·12H_2_O (analytical grade, CAS No. 15120-17-9) was used as the arsenic source in this study.

### 2.2. Experimental Design

The experiment was conducted in the greenhouse of the Research Center for Eco-Environmental Sciences at the Chinese Academy of Sciences. The temperatures in the greenhouse during the day and night were (25 ± 3) °C and (20 ± 2) °C, respectively, with natural light. White plastic flower pots with 1 kg of soil each were used for the experiment. Four maize seedlings were planted in each pot, which were thinned to two plants after one week. The experiment was designed with three factors: different levels of arsenic addition (As, 0, 25, and 50 mg·kg^−1^), iron addition (Fe, 0 and 50 mg·kg^−1^), and AMF treatment status (inoculated and non-inoculated with AMF, abbreviated as M and NM, respectively), resulting in a total of 10 treatments (As0-Fe0-M, As0-Fe0+M, As25-Fe0-M, As25-Fe0+M, As25-Fe50-M, As25-Fe50+M, As50-Fe0-M, As50-Fe0+M, As50-Fe50-M, and As50-Fe50+M, where +M and -M represent M and NM, respectively.) with four replications of each parameter measurement, totaling 40 pots. After adding arsenic and iron sources to the soil, the pots were allowed to equilibrate for two weeks before planting. Before the experiment, basic fertilizers were added to the mixed soil in the form of a solution. The types and amounts of basic fertilizers added were: NH_4_NO_3_, KH_2_PO_4_, K_2_SO_4_, CaCl_2_·2H_2_O, MgSO_4_·7H_2_O, MnSO_4_·H_2_O, CuSO_4_·5H_2_O, ZnSO_4_·7H_2_O, and (NH_4_)_6_Mo_7_O_24_·4H_2_O with N 60 mg·kg^−1^, P 30 mg·kg^−1^, K 67 mg·kg^−1^, Ca 20 mg·kg^−1^, Mg 4.5 mg·kg^−1^, Mn 0.92 mg·kg^−1^, Cu 0.54 mg·kg^−1^, Zn 1.24 mg·kg^−1^, and Mo 0.06 mg·kg^−1^. During the growth period, small amounts of nitrogen and potassium fertilizers were supplemented to ensure the normal growth of plants. Distilled water was added daily during the growth period, and the soil moisture content was maintained at about 80% of the field capacity using a weighing method.

### 2.3. Harvest and Analysis

#### 2.3.1. Sampling

After the maize plants had grown for 56 days, the roots, stems, and leaves were harvested and separated into three portions. The samples were cleaned by flushing with tap water and rinsing with deionized water before drying. Fresh root samples weighing between 0.6 g and 0.8 g were cut into approximately 1 cm segments and soaked in a centrifuge tube with 50% ethanol for preservation. The remaining parts of the plant, including the stem, were sterilized at 105 °C, dried in an oven at 75 °C for 72 h until constant weight was achieved, and then weighed and recorded for biomass determination. The samples were then powdered, extracted, and ground for further analysis. The leaf blades were stored in a refrigerator at −20 °C after removing the veins for subsequent analysis.

#### 2.3.2. Analysis and Determination

The determination of arsenic content in maize stems and roots was carried out using an Atomic Fluorescence Spectrophotometer (AFS-610, Beijing Ruili Analytical Instrument Company, Beijing, China, similarly hereinafter). The determination of phosphorus and iron content in plant boiling solution was carried out using an Inductively Coupled Plasma Optical Emission Spectrometer (ICP-OES, Optima 2000DV, Perkin Elmer Co., Waltham, Massachusetts, USA, similarly hereinafter). During boiling, standard samples provided by the National Standards Bureau (Shrub Leaves, GBW07603, GSV-2) were added to ensure the accuracy and precision of the boiling and analysis. To measure the amount of active iron in maize leaves, 0.3 g of a fresh leaf sample was combined with 3 mL of 1 M HCl. The mixture was oscillated at room temperature for 8 h at 9000 r·min^−1^, and then centrifuged for 10 min. The supernatant was filtered through a 0.45 μm membrane, and the iron concentration in the supernatant was determined using ICP-OES [39].

#### 2.3.3. Determination of Mycorrhizal Colonization Rate and Soil Fungal Hyphal Density

To determine the AMF colonization rate and root length of the root system, 1 cm root segment samples saved in centrifuge tubes were extracted and tested using the Trypan Blue—Grid Intersection Method [40,41]. After quenching, the remaining shoot and root parts were milled into powder and prepared for further analysis. The soil samples were carefully mixed, and duplicate subsamples of 2 g were blended at high speed in a Waring Blender with 250 mL of deionized water for 30 s. The soil samples with root pieces taken inside root compartments were shaken vigorously with 250 mL of water and then poured into the blender through a 710 μm sieve to retain the roots. The blended suspension was rapidly transferred to wide-necked flasks, agitated vigorously by hand shaking, and left on the bench for 60 s. Duplicate 5 mL aliquots were pipetted onto 25-mm Millipore filters. The fitters were covered with lactoglycerol-trypan blue for 5 min, rinsed with deionized water, and transferred to microscope slides to dry. Four filters from each original sample were mounted in lactoglycerol-trypan blue and intersections between blue-stained hyphae and a grid in the eyepiece were counted in 25 fields of view at 200× magnification and the fungal hyphal density was calculated [18,42].

#### 2.3.4. Determination of MDA Content and Antioxidant Enzyme Activities

Malondialdehyde (MDA) is an important product in the process of membrane lipid peroxidation, and its content reflects the strength of lipid peroxidation. Superoxide dismutase (SOD), peroxidase (POD), and catalase (CAT) are important antioxidant enzymes in plants that can scavenge reactive oxygen species produced in plants and avoid the potential damage of reactive oxygen species to plant cell membrane structure. The MDA content was determined using the thiobarbituric acid (TBA) method [43]. To prepare the tissue homogenate, 0.5 g of maize leaves were accurately weighed and ground in 5 mL of distilled water in a mortar until homogenized. The resulting homogenate was centrifuged at 4000 rpm for 15 min, and 1 mL of the supernatant was taken and made up to a final volume of 100 mL with distilled water in a volumetric flask. The activities of SOD and POD were measured using a colorimetric method [44]. The CAT activity was determined using the ammonium molybdate method [45]. To prepare the crude enzyme solution, 0.1 g of maize leaves was accurately weighed and ground in liquid nitrogen, and then 10 mL of 50 mmol·L^−1^ cold phosphoric acid buffer solution (pH 7.0) was added according to a mass-to-volume ratio. The resulting mixture was centrifuged at 10,000 rpm at 4 °C for 20 min, and the supernatant was collected as the crude enzyme solution. Finally, the MDA content (nmol·g^−1^ FW), POD and CAT activities (U·mg^−1^ protein·min^−1^), and SOD specific activity (U·mg^−1^ protein) were calculated using the formulae provided in the assay kits (Nanjing Jiancheng Bioengineering Institute, Nanjing, China).

#### 2.3.5. Determination of the Content of Soluble Protein and Non-Protein Thiol (NPT)

Soluble protein is one of the important components in the non-enzymatic antioxidant system, which can reduce the interference of heavy metal ions on the physiological and metabolic processes in plant cells. Protein concentrations were measured using a UV-6100 UV-Vis spectrophotometer. The addition of 0.1 M phosphate buffer (pH 7.0) was used to adjust the instrument to zero. Then, 2.9 mL of the 0.1 M phosphate buffer (pH 7.0) was taken and added to 0.1 mL of crude enzyme solution in a spectrophotometric cuvette with a light path of 1 cm. The mixture was thoroughly mixed and then the absorbance values at 280 nm and 260 nm wavelengths were measured. The results were then calculated and expressed in terms of protein concentration (mg·.mL^−1^) [44]. For NPT measurement, 0.3 g of fresh leaf samples were mixed with 3 mL of extraction solution (0.1 M HCl, 1 mM EDTA, 4% PVP) and homogenized. The mixture was then centrifuged for 10 min, and 0.3 mL of the supernatant was removed and mixed with 2.4 mL of buffer solution (0.12 M sodium phosphate, pH 7.8, 6 mM EDTA), followed by adding 0.3 mL of 6 mM DTNB and incubating for 30 min for color development. Finally, the absorbance was measured at 412 nm wavelength, a standard curve for GSH was prepared, and the NPT content was expressed in terms of nmol·g^−1^ fresh weight [46].

### 2.4. Data Statistical Analysis

SPSS software (version 20.0) was used for the data analysis. Data were graphed, and means and standard deviations were calculated using Microsoft Excel 2010 software. A three-way ANOVA was applied to examine the effects of AMF, As, and Fe and their interactions on shoot and root biomass, root length, mycorrhizal colonization, plant phosphorus, arsenic, and iron content, leaf membrane lipid peroxidation, soluble protein, non-protein thiol content, and enzymatic antioxidant activity of maize. A one-way ANOVA followed by a Duncan’s test was used to evaluate significant differences among different treatments at *p* < 0.05.

## 3. Results

### 3.1. Plant Growth and Mycorrhizal Establishment

The results (Table 1) showed that no mycorrhizal structures were detected in the roots of plants treated with NM, and the length density of fungal hyphae in the soil ranged from 0.17 to 0.42 m·g^−1^. In contrast, plants treated with AMF had root colonization rates of 46% to 61% and fungal hyphae densities of 1.31 to 4.13 m·g^−1^. Variance analysis showed that, regardless of the addition of iron, both the root colonization rate and soil fungal hyphae density of plants treated with AMF decreased as the arsenic addition level increased. The addition of iron increased the root colonization rate of plants treated with As25 and As50, respectively (Table 1).

As shown in Table 1, for Fe0 treatment, regardless of inoculation with AMF, the biomass of maize stems and roots, as well as root length, decreased with increasing arsenic addition levels. At the same arsenic addition level, compared with the NM treatment, the M treatment (AMF inoculation) significantly increased the biomass of maize stems and roots in the Fe0 treatment (*p* < 0.05), with increases of 31% and 20% under the As25 treatment and 105% and 25% under the As50 treatment, respectively. Among different arsenic addition treatments, compared with the Fe0 treatment, iron addition significantly increased the biomass of maize stems and roots in the As25-NM treatment (*p* < 0.05), with increases of 22% and 20%, respectively. At the same arsenic addition level, compared with the Fe0-NM treatment, Fe50+M treatment significantly increased the biomass of maize stems and roots in the As25 treatment, with increases of 53% and 35%, respectively, and in the As50 treatment, with increases of 123% and 32%, respectively. This treatment also significantly increased the root length of maize. Combination treatment with iron addition and AMF inoculation showed better effects on increasing the biomass and root length of maize stems and roots than single treatments.

### 3.2. Phosphorus, Arsenic, and Iron Contents in Plants

As shown in Figure 1, under the same level of arsenic addition, regardless of the addition of iron, M treatment significantly increased the phosphorus content in the maize stems and roots compared to the NM treatment (*p* < 0.05). In the As0 treatment, compared to the NM treatment, AMF inoculation significantly increased the phosphorus content in the stems and roots of maize by 62% and 91%, respectively. For the As25 treatment, AMF inoculation significantly increased the phosphorus content in the stems and roots of maize by 81% and 72%, respectively, under Fe0 treatment, and by 95.8% and 95.6%, respectively, under Fe50 treatment. For the As50 treatment, compared to the NM treatment, AMF inoculation significantly increased the phosphorus content in the stems and roots of maize by 123% and 122%, respectively, under Fe0 treatment, and by 87% and 111%, respectively, under Fe50 treatment.

As shown in Figure 2, inoculation with AMF or the addition of iron significantly reduced the arsenic content of maize roots under medium to low arsenic pollution concentrations. AMF inoculation also significantly reduced the arsenic content in maize stems. At the same arsenic addition level, the Fe50+M treatment showed a more significant reduction in both stem and root arsenic content compared with the Fe0-M treatment. Specifically, under As25 treatment, the reductions in arsenic content in stems and roots were 41% and 60%, respectively; under As50 treatment, the reductions were 33% and 37%, respectively. This suggests that the combination of inoculation with AMF and the addition of iron can more effectively reduce arsenic content in maize stems and roots than either treatment alone (Figure 2).

As shown in Table 2, the addition of arsenic significantly reduced the phosphorus-arsenic uptake ratio in both maize stems and roots in all treatments (*p* < 0.05). Under arsenic addition, whether or not iron was added, AMF inoculation significantly increased the phosphorus-arsenic uptake ratio in both stems and roots. At the same arsenic addition level, compared with the Fe0-M treatment, the Fe50+M treatment showed a greater increase in the phosphorus-arsenic uptake ratio in maize stems and roots.

As shown in Figure 3, the inoculation of AMF significantly increased the iron content in maize stems under As addition and significantly reduced the iron content in maize roots under Fe50-NM treatment. Irrespective of the addition of iron or inoculation with AMF, the As50 treatment resulted in a considerably greater increase in iron content in maize stems, albeit with a decreased iron content in roots compared to the As25 treatment.

### 3.3. Membrane Lipid Peroxidation and Active Iron Content in Maize Leaves

As shown in Figure 4, either with or without AMF treatment, the malondialdehyde (MDA) content in maize leaves significantly increased with the increase in arsenic (As) addition level (*p* < 0.05). At the same level of As addition, M treatment significantly reduced the MDA content in maize leaves compared to NM treatment at the same amount of iron, and iron treatment also significantly reduced the MDA content in maize leaves compared to Fe0 treatment under the same AMF treatment. Among the different levels of As addition, Fe50+M had a better effect on reducing the MDA content in maize leaves than Fe0-M treatment (*p* < 0.05) and showed significant reductions of 28% and 30% in As25 and A50 treatments, respectively (Figure 4).

For plants that received the Fe0 treatment, the content of active iron in maize leaves treated with NM showed a trend of first decreasing and then increasing with the increase in As addition level. The content of active iron in maize leaves treated with AMF increased with As addition level, and reached a significant level when the exogenous As in soil increased to 50 mg·kg^−1^ (*p* < 0.05). The addition of iron significantly increased the content of active iron in maize leaves in As50 and As25-M treatments, and AMF inoculation significantly reduced the content of active iron in the As25-Fe50 treatment (Figure 4).

### 3.4. Contents of Soluble Proteins and Non-Protein Thiols (NPT) in Maize Leaves

As shown in Figure 5, for the Fe0 treatment, the content of soluble proteins in maize leaves of NM-treated plants increased with the addition of arsenic and reached a significant level (*p* < 0.05) when the external arsenic level increased to 50 mg·kg^−1^. Under arsenic addition, the inoculation of AMF reduced the content of soluble proteins in maize leaves in Fe0 treatment compared to NM treatment. At the same level of arsenic addition, Fe50+M treatment significantly reduced the content of soluble proteins in maize leaves compared to Fe0-M treatment (*p* < 0.05), with significant decreases of 28% and 21% under As25 and As50 treatments, respectively.

For Fe0 treatment, the NPT content in maize leaves of NM-treated plants showed a trend of first increasing and then decreasing with the addition of arsenic, as shown in Figure 5. Under arsenic addition, M treatment significantly reduced the NPT content in maize leaves in Fe0 treatment compared to NM treatment, with decreases of 55% and 22% under As25 and As50 treatments, respectively. In addition, the addition of iron significantly reduced the NPT content in maize leaves of NM-treated plants compared to the Fe0 treatment, with decreases of 26% and 15% under the As25 and As50 treatments, respectively. Furthermore, Fe50+M treatment significantly reduced the NPT content in maize leaves compared to Fe0-M treatment under As25 treatment (*p* < 0.05) by 55%.

### 3.5. Enzymatic Antioxidants in Maize Leaves

The enzymatic antioxidant system induced by As stress in maize leaves mainly consists of CAT, POD, and SOD. The results (Figure 6) showed that for Fe0 treatment, the addition of As significantly increased CAT activity and significantly decreased POD and SOD activities in NM-treated maize leaves. Under As25 treatment, the CAT, POD, and SOD activities in Fe0-treated maize leaves with AMF inoculation showed no significant changes compared to those with NM treatment (*p* > 0.05). Under As50 treatment, the CAT activity was significantly decreased and the POD and SOD activities were significantly increased in Fe0-treated maize leaves with AMF inoculation compared to those with NM treatment. This indicates that AMF inoculation can induce POD and SOD activities in maize leaves under moderate As stress. At the same As level, the addition of iron resulted in no significant changes in CAT, POD and SOD activities in NM-treated maize leaves compared to Fe0 treatment (*p* > 0.05). As shown in Figure 6, under As25 treatment, neither AMF inoculation nor iron addition showed significant changes in CAT, POD, and SOD activities in Fe0-M-treated maize leaves, while Fe50+M treatment significantly increased CAT, POD, and SOD activities by 18%, 22%, and 36%, respectively, compared to Fe0-M treatment. Thus, the results show that the combined treatment of iron addition and AMF inoculation can significantly enhance antioxidant enzyme activities in maize leaves with low As contamination.

### 3.6. Correlation Analyses

The correlation analysis of maize stem biomass, As and P content, and leaf antioxidant physiology was carried out. Plant biomass was significantly negatively correlated with stem As content and leaf MDA content and significantly positively correlated with leaf SOD activity. In addition, stem As content was significantly positively correlated with leaf MDA content (Table 3). Therefore, arsenic pollution induces oxidative damage in maize leaves and reduces shoot biomass.

## 4. Discussion

### 4.1. Effects of AMF Inoculation and Iron Supplementation on Membrane Lipid Peroxidation in Maize Leaves

This study showed increased MDA content in maize leaves with higher levels of arsenic application (*p* < 0.05). Hartley-Whitaker et al. [10], however, tested the arsenic-resistant genotype of velvet grass and found no significant changes in membrane lipid peroxidation under different arsenic concentrations. In contrast, in the non-resistant genotype, exposure to arsenic rapidly increased membrane lipid peroxidation in plant roots, illustrating that the addition of arsenic causes the accumulation of active oxygen (ROS) and exacerbates lipid peroxidation in non-resistant plant cells. During normal plant growth, ROS production and removal are in a dynamic balance, and the low free radical concentration will not cause plant damage. However, under arsenic contamination stress, the reduction of pentavalent arsenic to trivalent arsenic in plant cells disrupts the balance of ROS production and elimination, leading to ROS accumulation [47,48]. When the accumulation of ROS exceeds a certain concentration, it causes the oxidation of unsaturated fatty acids in cell membrane lipids in a process known as membrane lipid peroxidation. This process produces MDA, and the MDA content reflects the degree of lipid peroxidation. In this work, AMF inoculation and iron supplementation both reduced the MDA content in maize leaves for plants grown in arsenic-contaminated soil. This is related to the addition of iron and the inoculation of AMF to reduce the accumulation of arsenic in maize shoots. After AMF treatment, specific sites in the extra-root mycelium and spores of AMF can bind to heavy metal ions in the soil, thus immobilizing them in the mycelium and limiting their transfer to the host plant [49]. This suggests that AMF can store the heavy metal ions absorbed from the soil in the fungal structure and thus reduce the extent of plant damage. The membrane lipid structure of plant cells is protected.

### 4.2. Effects of AMF Inoculation and Iron Supplementation on the Contents of NPT and Soluble Proteins in Maize Leaves

Phytochelatins (PCs) are a type of cysteine-rich peptide [50]. Heavy metal pollutants, such as copper, arsenic, cadmium, and zinc, can induce the synthesis of PCs [51,52] in plants that can act as detoxifying agents by chelating heavy metals with their thiol groups. Previous studies have shown that plant non-protein thiols (NPTs) are composed mainly of PCs, so they can be measured to indirectly reflect PC content [53]. In this study, the addition of arsenic increased the NPT content in maize leaves treated with NM, and AMF inoculation resulted in a trend of first decreasing and then increasing the NPT content in maize leaves with arsenic addition. Both AMF inoculation and iron supplementation reduced the NPT content in maize leaves treated with NM under low arsenic pollution. These results suggest that the addition of arsenic to soil may induce maize PC synthesis and thus increase the NPT content in leaves, with the medium arsenic treatment showing a relatively higher NPT content. However, the inoculation of AMF and iron supplementation significantly reduced the total arsenic content in maize plants under low arsenic pollution, alleviating arsenic toxicity and decreasing the NPT content in leaves.

Soluble protein is one of the important components of the non-enzymatic antioxidant system [54], participating in cell osmotic regulation and also directly binding to heavy metal ions through its abundant hydroxyl, carboxyl, amino, aldehyde, and phosphate groups. This binding can reduce the interference of heavy metal ions with physiological metabolic processes in cells and decrease the damage to plant macromolecules such as DNA to improve plant growth [55]. In this study, with the increasing addition of arsenic, the content of soluble protein in maize treated with the NM-Fe0 combination also increased. This suggests that arsenic stress may induce protein production, and AMF inoculation may weaken the antioxidant function of soluble protein by enhancing plant arsenic resistance, resulting in a decrease in the soluble protein content. AMF inoculation can alter the biomass and root structure of plants under conditions of heavy metal contamination. This affects the uptake and translocation of heavy metals by the host, explaining one way that AMF treatment induces plant detoxification [56].

### 4.3. Effects of Inoculating AMF and Iron Supplementation on Antioxidant Enzyme Activities in Maize Leaves

When plants are subjected to abiotic stress, excessive ROS accumulation can occur, leading to oxidative damage. SOD, POD, and CAT are important antioxidant enzymes in plants. Despite having low activity under normal growth conditions, these enzymes can scavenge ROS, thereby preventing damage to the plant cell membrane structure. Arsenic-induced oxidative stress results in ROS accumulation within the plant, triggering activation of the endogenous antioxidant enzyme system (including SOD, POD, and CAT) and the increased antioxidant enzyme activities [57] scavenge excess ROS and protect against oxidative damage. SOD mainly functions to eliminate O_2_^−^, generating non-toxic O_2_ and less toxic H_2_O_2_. CAT mainly scavenges H_2_O_2_ generated by oxidative enzymes such as superoxide dismutase (SOD), glycolate oxidase, and urate oxidase, and POD helps CAT remove excess H_2_O_2_ and other peroxides [58]. In this study, with the increase in arsenic level, the CAT activity in NM-Fe0-treated maize leaves increased, but at 50 mg·kg^−1^ of arsenic addition to the soil, both POD and SOD activities decreased significantly. The CAT activity in AMF-treated plants first increased and then decreased, indicating that the accumulation of arsenic may have caused oxidative stress and increased the activity of oxidases in the plants. However, the decrease in CAT activity in the As50+M treatment, and the decrease in POD activity and SOD specific activity in the As50-M treatment might reflect too much accumulated arsenic. This result is partly consistent with the conclusion of Mascher et al. [20] that SOD and POD activities in the aboveground part of Red Clover first increased and then decreased with increasing levels of arsenic. The results of this study show that under high arsenic stress, AMF inoculation can induce POD and SOD activities in maize leaves, which is consistent with the results reported by Zhan et al. [59]. In soil polluted with heavy metals, AMF can alleviate the toxicity of plants to heavy metals by enhancing the antioxidant defense capacity of leaves and the absorption of P by roots. Studies have shown that after AMF inoculation, plants reduce oxidative stress by enhancing the regulatory capacity of plant antioxidant systems and osmotic adjustment systems [60]. In addition, under arsenic addition, the combined treatment of iron supplementation and AMF inoculation can increase the CAT, POD, and SOD activities in maize leaves. This indicates that iron and AMF synergistically enhance the antioxidant defense system to improve the resistance of maize to arsenic pollution stress.

## 5. Conclusions

Under the stress of medium to low arsenic contamination, inoculation with arbuscular mycorrhizal fungi and the addition of iron compounds can synergistically promote phosphorus uptake while inhibiting arsenic uptake, leading to an increase in the phosphorus-arsenic uptake ratio and ultimately promoting maize growth. The synergistic effect of exogenous AMF and iron can alleviate lipid peroxidation in maize leaves under medium to low arsenic stress, reduce the soluble protein content and non-protein thiol content in low arsenic-polluted maize leaves, and enhance the activities of hydrogen peroxide, peroxidase, and superoxide dismutase in low arsenic-polluted maize leaves. Thus, exogenous AMF and iron compounds can synergistically promote the antioxidant defense system to counter arsenic toxicity in maize. In conclusion, AMF and iron compounds can be used to effectively improve plant tolerance to As by synergistically regulating plant physiological and biochemical mechanisms. The results of this work provide a theoretical basis for the remediation of medium to low arsenic-polluted agricultural soils.

## Figures and Tables

**Figure 1 jof-09-00677-f001:**
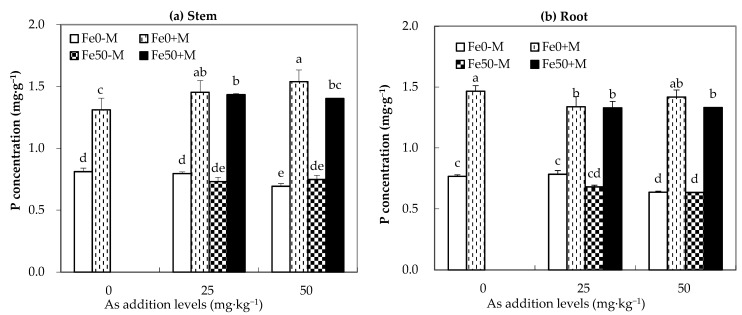
Phosphorus concentration (mg·g^−1^) in maize plants under different As, Fe addition levels, and AMF inoculation. (**a**,**b**) represent the phosphorus concentration of maize stems and roots, respectively. Fe0-M: indicates no iron and no inoculation AMF treatment; Fe0+M: indicates no iron addition and inoculation AMF treatment; Fe50-M: indicates iron addition and non-inoculation AMF treatment; Fe50+M: indicates iron addition and inoculation AMF treatment. Data are the means of four replicates (mean ± SEs). Different lowercase letters indicate significant differences between treatments at *p* < 0.05.

**Figure 2 jof-09-00677-f002:**
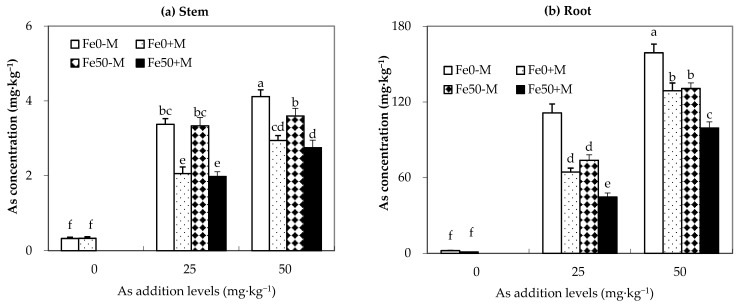
Arsenic concentration (mg·kg^−1^) in the maize plants under different As, Fe addition levels, and AMF inoculation. (**a**,**b**) represent the arsenic concentration of maize stems and roots, respectively. Fe0-M: no iron and no inoculation AMF treatment; Fe0+M: no iron addition and inoculation AMF treatment; Fe50-M: no iron addition and no inoculation AMF treatment; Fe50+M: iron addition and inoculation AMF treatment. Data are the means of four replicates (mean ± SEs). Different lowercase letters indicate significant differences between treatments at *p* < 0.05.

**Figure 3 jof-09-00677-f003:**
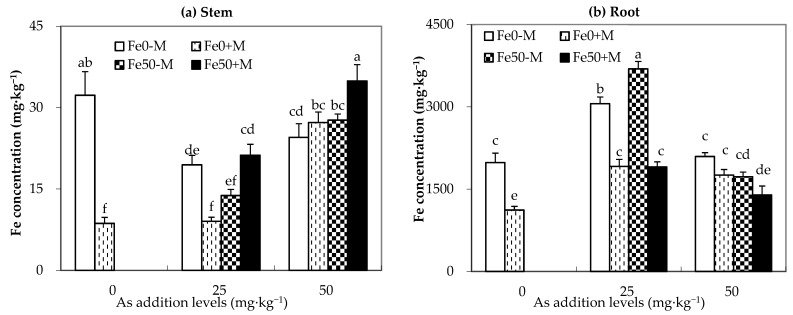
Iron concentration (mg·kg^−1^) in the maize plants under different As, Fe addition levels, and AMF inoculation. (**a**,**b**) represent the iron concentration in the stems and roots of maize, respectively. Fe0-M: no iron and no inoculation AMF treatment; Fe0+M: no iron addition and inoculation AMF treatment; Fe50-M: no iron addition and no inoculation AMF treatment; Fe50+M: iron addition and inoculation AMF treatment. Data are the means of four replicates (mean ± SEs). Different lowercase letters indicate significant differences between treatments at *p* < 0.05.

**Figure 4 jof-09-00677-f004:**
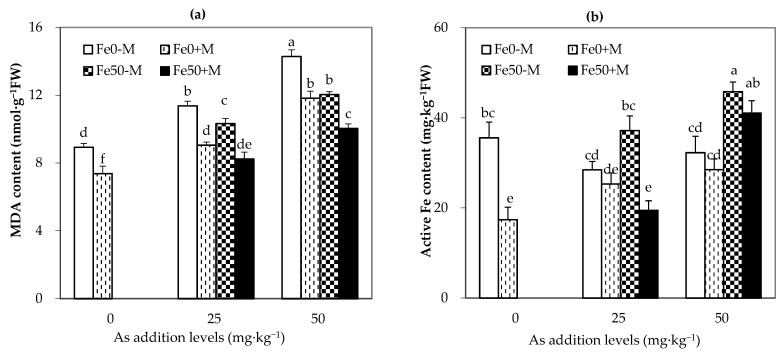
Malondialdehyde and active Fe concentrations in the leaves under different As, Fe addition and AMF inoculation. (**a**,**b**) represent MDA content and active iron concentration in maize leaves, respectively. Fe0-M: no iron and no inoculation AMF treatment; Fe0+M: no iron addition and inoculation AMF treatment; Fe50-M: no iron addition and no inoculation AMF treatment; Fe50+M: iron addition and inoculation AMF treatment. Data are the means of four replicates (mean ± SEs). Different lowercase letters indicate significant differences between treatments at *p* < 0.05.

**Figure 5 jof-09-00677-f005:**
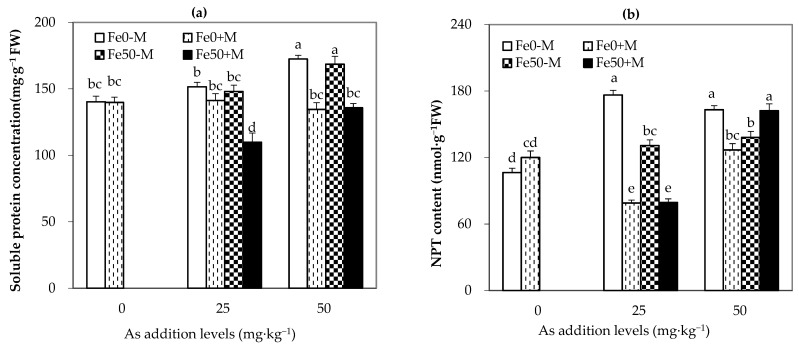
Non-enzymatic antioxidants in the leaves of maize plants under different As, Fe addition and AMF inoculation. (**a**,**b**) represent soluble protein content and non-protein sulfhydryl (NPT) content in maize leaves, respectively. Fe0-M: no iron, no inoculation AMF treatment; Fe0+M: no iron, inoculation AMF treatment; Fe50-M: no iron, no inoculation AMF treatment; Fe50+M: iron addition, inoculation AMF treatment. Data are the means of four replicates (mean ± SEs). Different lowercase letters show significant differences between treatments at *p* < 0.05.

**Figure 6 jof-09-00677-f006:**
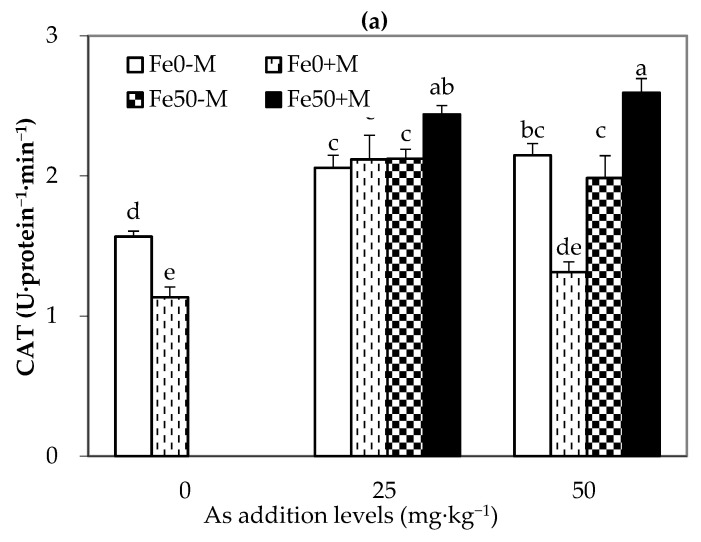
Enzymatic antioxidants in the leaves of maize plants under different As, Fe addition levels and AMF inoculation. (**a**–**c**) represent catalase (CAT) activity, peroxidase (POD) activity and total superoxide dismutase (SOD) activity in maize leaves. Fe0-M: no iron and no inoculation AMF treatment; Fe0+M: no iron addition and inoculation AMF treatment; Fe50-M: no iron addition and no inoculation AMF treatment; Fe50+M: iron addition and inoculation AMF treatment. Data are the means of four replicates (mean ± SEs). Different lowercase letters indicate significant differences between treatments at *p* < 0.05.

**Table 1 jof-09-00677-t001:** Maize and fungus growth under different Fe addition levels and AMF inoculation.

As Levels (mg·kg^−1^)	Fe Levels (mg·kg^−1^)	Inoculation Status	Biomass (g·pot^−1^)		Mycorrhizal Colonization Rate (%)	Root Length (m·pot^−1^)	Hyphae Length Density (m·g^−1^)
			Stem Dry Weight	Root Dry Weight			
As0	Fe0	NM	1.55 ± 0.06 c	3.42 ± 0.08 a	0	380.5 ± 16.4 a	0.35 ± 0.02 d
		M	1.96 ± 0.05 a	3.60 ± 0.10 a	60.3 ± 4.4 a	342.1 ± 30.5 a	3.27 ± 0.34 a
As25	Fe0	NM	1.15 ± 0.06 f	2.60 ± 0.06 c	0	252.1 ± 9.4 c	0.34 ± 0.02 d
		M	1.51 ± 0.06 cd	3.13 ± 0.07 b	49.5 ± 0.7 b	225.6 ± 7.6 cd	2.43 ± 0.16 b
	Fe50	NM	1.40 ± 0.03 cde	3.12 ± 0.10 b	0	268.0 ± 19.4 bc	0.39 ± 0.02 d
		M	1.77 ± 0.05 b	3.51 ± 0.06 a	58.4 ± 1.4 a	278.7 ± 20.9 b	2.70 ± 0.08 b
As50	Fe0	NM	0.62 ± 0.04 g	1.99 ± 0.08 d	0	180.5 ± 4.4 de	0.41 ± 0.02 d
		M	1.27 ± 0.06 ef	2.49 ± 0.10 c	46.3 ± 0.4 b	192.9 ± 13.8 de	1.79 ± 0.19 c
	Fe50	NM	0.55 ± 0.02 g	1.86 ± 0.07 d	0	165.1 ± 7.2 e	0.35 ± 0.03 d
		M	1.38 ± 0.06 de	2.63 ± 0.08 c	49.3 ± 1.5 b	282.0 ± 11.8 b	1.95 ± 0.12 c

Data are the means of four replicates (mean ± SEs). Different lowercase letters show significant differences between treatments at *p* < 0.05.

**Table 2 jof-09-00677-t002:** Uptake ratio of P to As under different As, Fe addition levels, and AMF inoculation.

As Levels (mg·kg^−1^)	Fe Levels (mg·kg^−1^)	Inoculation Status	Uptake Ratio of P to As	
			Stem	Root
As0	Fe0	NM	2605.2 ± 378.43 b	368.1 ± 22.75 b
		M	4206.3 ± 630.30 a	1479.6 ± 49.55 a
As25	Fe0	NM	237.0 ± 9.99 c	7.2 ± 0.71 c
		M	716.5 ± 46.9 c	20.9 ± 1.74 c
	Fe50	NM	222.0 ± 15.5 c	9.3 ± 0.62 c
		M	730.5 ± 41.01 c	30.2 ± 1.9 c
As50	Fe0	NM	169.1 ± 7.63 c	4.0 ± 0.11 c
		M	504.2 ± 12.56 c	11.0 ± 0.96 c
	Fe50	NM	208.8 ± 9.89 c	4.9 ± 0.21 c
		M	518.9 ± 41.04 c	13.5 ± 0.94 c

Data are the means of four replicates (mean ± SEs). Different lowercase letters indicate significant differences between treatments at *p* < 0.05.

**Table 3 jof-09-00677-t003:** The correlation coefficient of stem biomass, As, P content and leaf antioxidant physiology in maize.

	Biomass	As	P	MDA	NPT	CAT	POD
As	−0.806 **						
P	0.588	−0.356					
MDA	−0.923 **	0.855 **	−0.495				
NPT	−0.584	0.586	−0.483	0.659 *			
CAT	−0.244	0.498	−0.081	0.178	0.130		
POD	0.537	−0.374	0.586	−0.407	−0.498	−0.043	
SOD	0.725 *	−0.451	0.691 *	−0.620	−0.572	0.095	0.897 **

** Significant difference (*p* < 0.01); * significant difference (*p* < 0.05).

## Data Availability

Not applicable.

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
