# Peer review of "Synergistic Reduction of Arsenic Uptake and Alleviation of Leaf Arsenic Toxicity in Maize (Zea mays L.) by Arbuscular Mycorrhizal Fungi (AMF) and Exogenous Iron through Antioxidant Activity"

_jof, 2023, doi:10.3390/jof9060677_

Round 1
Reviewer 1 Report
The manuscript entitled “Synergistic Reduction of Arsenic Uptake and Alleviation of Leaf Arsenic Toxicity in Maize (Zea mays L.) by Arbuscular Mycorrhizal Fungi (AMF) and Exogenous Iron through Anti-oxidant Activity” aimed to study the synergistic antioxidant mechanisms of AMF (Funneliformis mosseae) and Fe compounds in alleviation of As toxicity in maize (Zea mays L.) growing under low and moderate As contamination.
The study some important results which demonstrated that the co-inoculation of AMF and Fe compound addition significantly reduced the As concentration in stem and root, malondialdehyde (MDA) content in leaf, and soluble protein and non-protein thiol (NPT) contents in leaf of maize under As25 and As50 treatments. In addition, co-inoculation of AMF and Fe compound addition significantly increased the activities of catalase (CAT), peroxidase (POD), and superoxide dismutase (SOD) in the leaves of maize under As25 treatment. In conclusion, the results indicated that the co-inoculation of AMF and Fe compound addition can inhibit As uptake and promote P uptake by maize under low and moderate As contamination, thereby mitigating the lipid peroxidation on maize leaves and reducing As toxicity by enhancing the activities of antioxidant enzymes under low As contamination.
Overall, the study is well-conducted and revealed significant findings that should be taken intro consideration to improve maize growth and tolerance to As. However, I have some revisions that should be taken into consideration as shown below;
- The introduction section should include the clear hypothesis of the present study as well as the objective should be written in more details.
- The number of replications of each parameter measured should be mentioned in the methods section.
Measurements should be written in more details so that they can be replicated easily
-The Results section is well explained.
- The discussion should be improved and discussed the findings in details including the explanations and comparison to literature.
- References should be updated as most of the references included are not recent enough.
Minor editing of English language is required
Reviewer 2 Report
Comments to the manuscript: Synergistic Reduction of Arsenic Uptake and Alleviation of Leaf Arsenic Toxicity in Maize (Zea mays L.) by Arbuscular Mycorrhizal Fungi (AMF) and Exogenous Iron through Anti- oxidant Activity
General comments:
This study aims to elucidate the physiological and biochemical mechanisms of how AMF inoculation and exogenous iron compounds mitigate the phytotoxic effects of arsenic contamination. The authors should clearly state how they reach to understand how AMF mitigate effects of arsenic and the physiological and biochemical mechanisms involved. I think that it is necessary a hypothesis of this study.
Introduction
There are no hypotheses in the introduction of this study, I recommend including it at the end of this section.
Methods
Lines 199-202, please give more detail of this procedure
Line 204, please define what does MDA means, since is the first time the authors use this acronym in this section.
Please give more information of the analysis carried on the data statistical analysis section, for example what ANOVA were employed, why they used Duncan test, if data were transformed, etc.
I don't see an analysis that would allow the authors to solve the aim of explain how the physiological and biochemical mechanisms mitigate the phytotoxic effects of arsenic.
Results
The authors should explain if hyphae founded on NM treatments are AMF´s hyphae, and if the answer is yes, why they founded AMF´s hyphae in the NM treatment.
Lines 269-270 please correct “AMF significantly increased the phosphorus content in the stems and roots of maize treated with NM”. I assume that the authors mean M treatment.
Discussion
The authors must highlight how the aims of the work were approached.
Lines 396-402, this is not a discussion, the authors should move this information to introduction section or use it at the end of this paragraph to explain their results.
Conclusions
The authors should mention if the aim of elucidating the physiological and biochemical mechanisms of AMF inoculation and exogenous iron compounds mitigate the phytotoxic effects of arsenic contamination was reached.
Reviewer 3 Report
Dear Authors,
Please rewrite between +M and -M in the experimental design. Does - M stand for NM?
Authors will need to be consistency for using maize or corn in their writing, for example line 250 and line 270, line 287?
repeated texts appeared in line 125-146 and 148-168.
Maize normally harvest at 75-110 days? what was the reason for this experiment to end at 56 days?
rate of infection, how many replicates were done? no SE?
NA
Reviewer 4 Report
Dear authors,
The paper entitled "Synergistic Reduction of Arsenic Uptake and Alleviation of Leaf Arsenic Toxicity in Maize (Zea mays L.) by Arbuscular Mycorrhizal Fungi (AMF) and Exogenous Iron through Anti-oxidant Activity" shows a valid theme and the results of this type of study can serve to complement information regarding bioremediation of contaminated soil. I think that the results could be better presented, maybe using more updated graphism and I think that lacks a deeper discussion. Also, since you are trying to publish in the Journal of Fungi, I think that the AMF aspect should be further explored,
Nevertheless, I have some suggestions in the attached PDF.

Some editing of English language is required.
Round 2
Reviewer 1 Report
The revised version is improved.
Author Response
Dear Reviewer
The author again revised and improved the previous opinions. For example, the author discussed the role of AMF in explaining the relevant results of this study and added the latest references. In addition,The author added ' 3.6 correlation analysis ' in the result analysis part to analyze the correlation between stem biomass, As, P content and leaf antioxidant physiology of maize.
Reviewer 4 Report
Dear authors,
I appreciate the revision, but I still think that the results could be better presented, maybe using more updated graphism and I think that lacks a deeper discussion. Also, I insist on the issue that you are trying to publish in the Journal of Fungi, and the AMF role should be further explored.
Dear Editor,
I appreciate the opportunity to review the article again; all my comments are provided above. Although I find the topic essential and very interesting, I find the article still lacks novelty. Also, I think that the AMF role should be further explored, and was not. Finally, I still think that a lot of the language used in this article is very similar to the previous publication "Mycorrhiza and Iron Tailings Synergistically Enhance Maize Resistance to Arsenic on Medium Arsenic‑Polluted Soils Through Increasing Phosphorus and Iron Uptake".
My opinion is that some of the minor problems were corrected, but my main comments were not.
Best regards,
Carla
